# GMML IS ALL YOU NEED

## ABSTRACT

Vision transformers have generated significant interest in the computer vision community because of their flexibility in exploiting contextual information, whether it is sharply confined local, or long range global. However, they are known to be data hungry. This has motivated the research in self-supervised transformer pretraining, which does not need to decode the semantic information conveyed by labels to link it to the image properties, but rather focuses directly on extracting a concise representation of the image data that reflects the notion of similarity, and is invariant to nuisance factors. The key vehicle for the self-learning process used by the majority of self-learning methods is the generation of multiple views of the training data and the creation of pretext tasks which use these views to define the notion of image similarity and data integrity. However, this approach lacks the natural propensity to extract contextual information. We propose Group Masked Model Learning (GMML), a Self-Supervised Learning (SSL) mechanism for pretraining vision transformers with the ability to extract the contextual information present in all the concepts in an image. This is achieved by manipulating randomly groups of connected tokens, ensuingly covering a meaningful part of a semantic concept, and then recovering the hidden semantic information from the visible part of the concept. GMML implicitly introduces a novel data augmentation process. Unlike most of the existing SSL approaches, GMML does not require momentum encoder, nor rely on careful implementation details such as large batches and gradient stopping, which are all artefacts of most of the current SSL techniques. Since its conception at the beginning of 2021, GMML maintains itself as unbeaten SSL method with several desirable benefits and marked a significant milestone in computer vision by being one of the first self-supervised pretraining methods which outperform supervised pretraining consistently with a large margin. GMML is currently the best mechanism to extract information from a given dataset and instil this information into transformer's weights. The code will be made publicly available for the community to train on bigger corpora.

**Impact of GMML:** We proposed GMML for self-supervised learning of vision transformers at the beginning of 2021 using masked autoencoder with reconstruction loss, however the idea is generally applicable to other losses [1, 2, 3]. The merits of GMML were shown employing small models and small/medium scale datasets due to extremely restricted computational resources. Since then, GMML has been widely adopted in computer vision and other related fields. Towards the end of 2021, SIMMIM [4] and MAE [5] extended GMML with reconstruction loss using huge vision transformers on large scale datasets, like ImageNet-1K [6]. GMML is now the leading SSL framework on multiple application areas, giving sate-of-the-art results for image classification [2], segmentation [4], audio analysis [7], medical image analysis [8, 9], video representation [10], etc.

# 1 INTRODUCTION

Vision transformers (ViT) [11] have shown tremendous potential due to self-attention mechanism which is able to model global context. Borrowing idea from Natural Language Processing (NLP) [12, 13] the ViT also treat an image as 1D sequence of visual tokens. This induces lack of intrinsic inductive bias to model local visual structure. Therefore, ViT requires orders of magnitude more data to model this inductive bias [11]. Very recently, vision transformers have been shown to perform well on ImageNet-1K [6] without external data [14]. However, they need distillation approaches and guidance from CNNs counterparts. Another hindrance preventing a wide spread

adoption of vision transformers (ViTs) is their tremendous computational demand [11] despite the improvements in vision transformers architecture design [15, 16]. These drawback particularly affect AI researchers with a smaller resource budget.

Self-supervised pretraining (SSP) can be an alternative to data hungry supervised pretraining (SP) of the ViTs. SSP of transformers is the defacto standard for natural language processing (NLP) [13] due to its success. However, SP is still the default due to its superiority over SSP. A tremendous progress in SSL for visual data has been marked by recent methods [17, 18, 19, 20] prior to GMML. A common theme to these non-GMML based methods is the learning of invariant representations for different views (distortions/augmentations) of the visual data by maximising the similarity between these different views. However, most of these approaches suffer from trivial constant solutions. To avoid trivial solution, these SSL approaches rely on careful implementation details such as large batches, gradient stopping, weight updates by moving average, asymmetric projection head, etc. In contrast to existing unsupervised learning approaches, GMML exploits information redundancy and complementarity in the image data by learning to reconstruct local contentby linking it to context. In spirit, this principle is similar to the masked language modelling (MLM) used in BERT [13] which recover masked words from context. The principle of predicting words from context is also inspired from word2vec [21]. In computer vision, we take the inspiration from the principle of denoising autoencoder [22] and from the idea of context encoder [23] which has been studied for unsupervised learning using CNNs. The main aim of this study is to merely extended the principles of MLM, denoising autoencoders, and context encoders to vision transformers for self-supervised learning. This is achieved by three principles: i) *learning to reconstruct the input stimulus by a mechanism akin to autoencoding, implemented by means of random data perturbation using masking of groups of connected tokens, etc.* ii) *a perception-action mechanism* [24], *which learns to recognise an action from its impact on perception*, and iii) *learning the notion of similarity of content from the preservation of content identity in the data.* The proposed SSL approach is instrumental in extracting an intrinsic data model and is admirably able to adapt to downstream tasks by fine tuning.

The GMML addresses the issues of data-efficiency of ViT by investigating how to train vision transformers from scratch, using limited data, by means of self-supervised pretraining, without using any external data. The proposed methodology of transformer pretraining by self-supervision is expected to have a significant impact on the advancement of science by enabling the wider research community starved of resources to contribute to deep learning. The main contributions and findings of this study are summarised as follows:

- We introduce GMML, a simple method for SSL of visual representations using transformer inspired from MLM of BERT, denoising autoencoder and context encoders.
- We endow the GMML architecture with a decoder and demonstrate that it can be implemented by essentially a couple of pointwise convolutional (linear) layers, thanks to the intrinsic characteristics of the transformer. This transformer based autoencoder avoids the need for a whole decoder block which is typically present in CNNs based encoder-decoder.
- The amount of labelled training data required for finetuning to learn a downstream task is two orders of magnitude lower than the supervised pretraining and finetuning.
- Total amount of training data (labelled and unlabelled) is also orders of magnitude lower.
- GMML outperforms state-of-the-art supervised/self-supervised methods in small, medium and large datasets with large margins reaching +35% improvement.
- To best of our knowledge for computer vision GMML is one of the first self-supervised pretraining method which outperformed supervised pretraining. We hope that this will enable the vision transformers to enjoy same success as BERT in NLP.
- GMML is among concurrent SSL methods which neither suffer from trivial solutions nor require careful implementation details, others being Barlowtwins [20] and VICReg [25].

## 2 METHOD

Unlike recent SSL joint embedding based methods [17, 18, 19, 20, 25, 26, 27, 28], GMML does not rely on maximising similarity between joint embeddings of different views of the image. Instead, GMML is motivated by a successful NLP pretext task MLM [13], denoising autoencoder [22] and context encoder [23] in image domain. There are several considerations when designing MLM

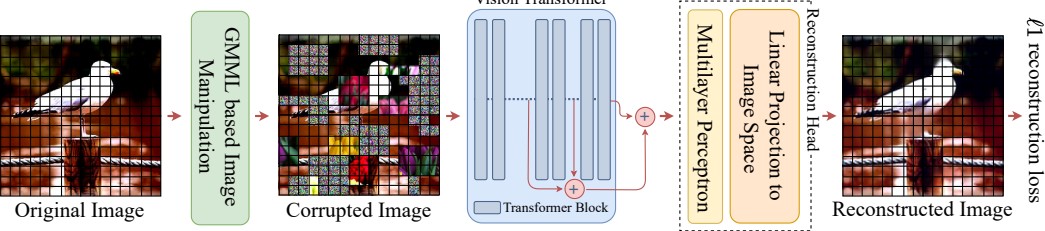

Figure 1: Group Mask Model Learning (GMML): The idea of GMML is to select random locations and randomly defined group of connected tokens around it to be corrupted by alien concept of choice, e.g., masking with zeros, random noise, or structured alien concept. Thus the GMML has both corrupted and uncorrupted group of connected tokens randomly selected defining a meaningful area of the image. We employ simple/light decoder with a couple of point-wise convolutions and simple loss like $\ell_1$ or $\ell_2$ reconstruction. We introduce skip connections from several intermediate transformer blocks to the decoder to get enriched representations from multiple levels.

alternative, masked image modelling (MIM), for image domain. These considerations are discussed in this section. The system diagram of GMML is shown in Figure 1.

## 2.1 CONSTRUCTION OF GMML

**Journey from MLM to GMML/MIM:** Data-tokens in NLP, i.e. words, most often represent semantic concepts. Consequently, randomly masking a small percentage of tokens and recovering them from context in NLP can induce semantic understanding in the transformer. On the contrary, individual data-tokens in an image, i.e. small visual patches, often do not represent a semantic concept. Therefore, randomly masking a small percentage of tokens is not as fruitful as in NLP.

Instead, we propose to randomly mask groups of connected tokens. These randomly defined groups of connected tokens are more likely to represent meaningful parts of different semantic concepts in an image. Hence, recovering these meaningful parts from the local and global contextual semantic information can induce learning of higher level concepts in the vision transformers. We note that the groups of randomly masked tokens will fall on different semantic concepts present in the image.

The Key hypothesis is that if the transformers are able to model missing information from groups of masked tokens on different objects, then they will implicitly learn the semantic representations of these objects in the image. This form the basis of the thesis that GMML is able to learn information from all the concepts. We refer to this mechanism of modelling missing information from groups of masked tokens as group mask model learning (GMML). The intuition is that by modelling all semantic concepts, GMML-based transformer will generalise better for unseen tasks, whether they are related to an object, a distributed object, or to the whole visual signal.

**Realisation of GMML via autoencoder:** The next question is to model learning of transformer weights via some self-supervised loss function. To realise the generic concept of GMML into a specific instance, the key idea of transformer based masked autoencoder was proposed. Although the idea of GMML is generic, we will mainly discuss the evolution of masked autoencoder via GMML.

The words in NLP have unique indices in the vocabulary (which correspond to class indices), hence, cross-entropy can be used to calculate the loss corresponding to masked tokens and update the network. However, due to the continuous nature of visual signal, there is no unambiguous notion of classes corresponding to patches which can be used for SSL. Therefore, the cross-entropy loss cannot be used out of the box to pretrain GMML based self-supervised vision transformers. One option is to extract representations of patches from a pretrained network and then cluster them into $k$ clusters. The cluster index can represent the class index for each patch enabling the use of cross entropy loss to recover masked patches. Another option can be quantisation of colour space to define classes, hence, enabling the use of cross entropy for recovering masked tokens. Another approach investigated by ViT [11] is to recover the average colour of the masked patch by masking a small percentage of patches randomly. These kind of approaches will inherit issues of visual vocabulary,

like, number of visual words in vocabulary, the quantisation error, visual ambiguity when assigning to cluster centres, etc.

Instead of following these lines for masked model learning, we prefer autoencoder based reconstruction loss. More specifically we use $\ell_1$-loss between the reconstructed image from GMML manipulated images and the original image. The reconstruction loss does not have the aforementioned issues associated with the quantisation-based approaches. The reconstruction loss suits more the continuous nature of the data by covering the dynamic range rather then quantising it. Beside, the reconstruction loss has the advantage of end-to-end self-supervised trainable system.

The proposed masked autoencoder has key differences from the vanilla autoencoders which are used commonly in CV. The existing autoencoder consists of usually convolutional encoders with non-linearity and pooling operations for downsampling a bottleneck representation and a decoder which consists of transposed convolutions or upsampling and convolutions. These decoders are usually expensive in terms of parameters as well as storage of feature maps. Due to the isotropic architecture design of vision transformers and their ability to exploit contextual local or global information, we employ a very light decoder. Our decoder consisting of two point-wise convolution layers (aka MLP layers in transformers) with ReLU non-linearity and a transposed convolution layer to return back to image space. Since the GMML architecture including both the transformer blocks as well as point-wise convolution is isotropic, therefore, some of the transformer blocks may act as decoder.

| Original Image | Alien Concept | Rec. Block 1 | Rec. Block 2 | Rec. Block 4 | Rec. Block 6 | Rec. Block 8 | Rec. Block 10 | Rec. Block 12 |

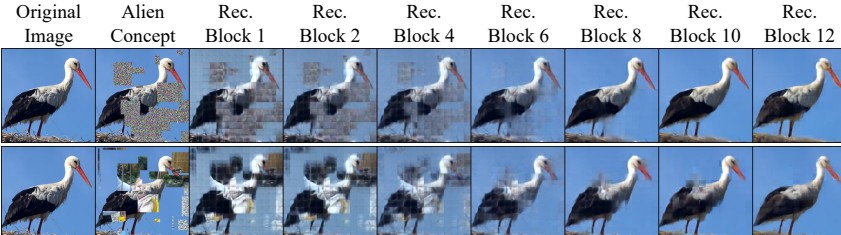

Figure 2: Reconstructed images from several transformer blocks after GMML pre-training employing Noise and Replace alien concepts. We only corrupt 30% of the images for demonstration purposes.

**Working insights:** There can be several variations to manipulate the images using GMML. Some are basic, others introduce the notion of alien concepts. The GMML enforces that initial blocks of the transformers have to model the alien concepts and then gradually defuse the native concepts present in the image via a mechanism similar to diffusion of information as shown in Figure 2 (refer to Appendix D for further visualisations and analysis). In the process, the transformer will gain the understanding of the concepts present in the image. In depth analysis of encoder and decoder separation in transformers blocks, detailed analysis of roles of transformers blocks at different stages, and the visualisation of transformers blocks are not the main focus of the paper and will be presented in a later study. The baseline is to pretrain the transformer based autoencoder without any masking. We noticed that the transformers are able to reconstruct the images perfectly after only few epochs. More importantly transformer need only first couple of blocks to do so, therefore, the rest of transformer blocks are mapping the identity. During the finetuning stage, the transformers obtained marginally better results compared to initialisation starting from random initialisation. We attributed this to the fact that without proper choice of constraints, i.e., mechanism of group mask model learning, autoencoders are unable to learn semantic concepts.

## 2.2 Choice of GMML based Image Manipulation

The amount and variation of alien concepts have different impact on performance. These choices are discussed below.

i) **Masking with zeros:** The most straightforward approach to introduce an alien concept is to mask the groups of connected tokens with zeros. We found empirically that it works well, however, it is not the most effective way as it is less difficult for the GMML to model the region masked with zeros.

ii) **Masking with noise:** A slightly more complex alien concept is to mask the connected tokens with random noise. Empirically, it works marginally better than masking with zeros. It is slightly more challenging for the network to model noise based alien concept.

iii) **Replace with visually plausible alien concept:** Most interesting manipulation is to introduce alien concepts from another image in the batch, instead of masking group of connected tokens with noise. This manipulation challenges the transformers as the network has to first model the visually plausible alien concepts which are injected at random locations via GMML. After modelling the visually plausible alien concepts in initial blocks the transformers use few blocks to gradually diffuse the information from the native concepts into the regions distorted by alien concepts by taking into account the local and global context of the image. Specifically, when introducing the plausible alien concept into an image we select a random image from the batch which can act as a negative image and use all the injected plausible alien concept from the same image. This may induce further regularisation into initial blocks of transformers when modelling the alien concepts which are coming from the same environment/context. Due to the introduction, modelling, and recovery from visually plausible alien concept, the network learns more meaningful semantic information.

iv) **Combinations of alien concepts:** We can combine any of the three strategies of GMML based manipulation described above. Refer to Section 3.2 for a detailed ablation study about the effect of the choice of align concepts.

## 2.3 ARCHITECTURE

Vision Transformer [11] receives as input a sequence of patches obtained by tokenizing the input image $\mathbf{x} \in \mathbb{R}^{H \times W \times C}$ into $n$ flattened $2D$ patches of size $p \times p \times C$ pixels, where $H$, $W$, and $C$ are the height, width, and number of channels of the input image and $n$ is the total number of patches. Each patch is then projected with a linear layer to $d$ hidden dimensions. In order to retain the relative spatial relation between the patches, a learnable position embeddings is added to the patch embeddings as an input to the transformer encoder. The transformer encoder $E(.)$ consists of $L$ consecutive multi-head self-attention (MSA) and multi-layer perceptron (MLP) blocks.

The objective of the image reconstruction is to restore the original image $\mathbf{x}$ from the GMML manipulated image $\hat{\mathbf{x}}$. For this task, we use the $\ell1$-loss between the original and the reconstructed image as shown in Equation 1. Although, $\ell2$-loss generally converges faster than $\ell1$-loss, $\ell2$-loss is prone to over-smooth the edges for image restoration [29]. Therefore, $\ell1$-loss is commonly used for image-to-image processing more than $\ell2$-loss.

$$\mathcal{L}(\mathbf{W}) = \sum_{k}^{N} \left( \sum_{i}^{H} \sum_{j}^{W} \mathbf{M}_{i,j}^{k} \times |\mathbf{x}_{i,j}^{k} - \bar{\mathbf{x}}_{i,j}^{k}| \right), \qquad \mathbf{M}_{i,j} = \begin{cases} 1, & \text{if } \mathbf{x}_{i,j} \text{ is manipulated} \\ 0, & \text{otherwise} \end{cases} \quad (1)$$

Where $\mathbf{W}$ denotes the parameters to be learned during training, $N$ is the batch size, $\mathbf{M}$ is a binary mask with 1 indicating the manipulated pixels, and $\bar{\mathbf{x}}$ is the reconstructed image.

To improve the performance of the autoencoder, we introduce skip connections from several intermediate transformer blocks to the decoder. These additional connections can directly send the feature maps from the earlier layers of the transformers to the decoder which helps to use fine-grained details learned in the early layers to construct the image. Besides, skip connections in general make the loss landscape smoother which is leading to faster convergence. Following, the reconstructed image $\bar{\mathbf{x}}$ is obtained by averaging the output features from intermediate blocks from the transformer encoder $E(.)$ and feeding the output to a light decoder $D(.)$ as follows: $\bar{\mathbf{x}} = D \left( \sum_{b \in \mathcal{B}} E_b(\hat{\mathbf{x}}) \right)$, where $E_b(.)$ is the output features from block $b$ and $\mathcal{B}$ is a pre-defined index set of transformer blocks that are included in the decoding process. In this work, we set $\mathcal{B}$ to $\{6, 8, 10, 12\}$.

## 3 EXPERIMENTAL RESULTS

The common evaluation to demonstrate the generalisation of the learnt features by self-supervised methods is to pretrain the model in unsupervised fashion, followed by fine-tuning the model on a downstream task like image classification, object detection, segmentation, etc. In this work, we conduct several experiments on 6 well-known multi-class datasets to show the effectiveness of our

proposed self-supervised vision transformer. In Section 3.1, the evaluation metrics and the results of the proposed method are explained. Next, we conduct several ablation studies to investigate the effect of the different recipes of the proposed approach in Section 3.2. We provide the details of the employed datasets and the full implementation details to train the proposed GMML model in a self-supervised fashion in Appendix C.

Table 1: Comparison with the SSL state-of-the-art methods. Both pretraining and fine-tuning are performed on the target dataset. * is reported by IDMM [30].

| Method | # pretrain epochs | Backbone | # params | Dataset | | | | |
|---|---|---|---|---|---|---|---|---|
| | | | | Flowers | Pets | CUB | Aircraft | Cars |
| *random init.* * | | | | 58.1 | 31.8 | 23.8 | 14.6 | 12.3 |
| SimCLR* [17] | | | | 71.1 | 52.1 | 36.2 | 43.2 | 64.3 |
| SupCon* [31] | | | | 72.3 | 50.3 | 37.8 | 29.4 | 66.2 |
| MoCov2* [32] | | | | 61.8 | 41.5 | 31.6 | 37.7 | 44.0 |
| MoCov3* [18] | 800 | ViT-T | 5M | 67.0 | 52.9 | 20.5 | 32.0 | 53.7 |
| DINO* [28] | | | | 64.1 | 51.3 | 41.8 | 45.7 | 65.3 |
| IDMM* [30] | | | | 79.9 | 56.7 | 43.1 | 43.2 | 66.4 |
| GMML (ours) | | | | 81.2 | 74.1 | 66.3 | 78.4 | 90.1 |
| GMML (ours) | 3000 | | | 90.4 | 86.0 | 71.2 | 84.1 | 92.7 |
| MAE [5] | 6000 | ViT-S | 22M | 86.9 | 73.0 | 59.4 | 69.0 | 91.0 |
| GMML (ours) | 3000 | | | **94.5** | **88.1** | **77.4** | **84.5** | **93.1** |

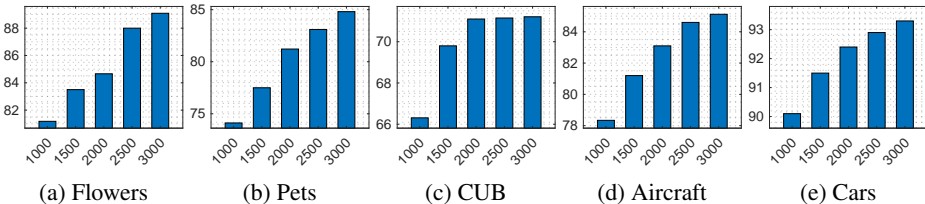

| (a) Flowers | (b) Pets | (c) CUB | (d) Aircraft | (e) Cars |
|---|---|---|---|---|

Figure 3: Effect of longer pretraining on small datasets employing ViT-T variant of transformers. The x-axis represents the pretraining epochs and y-axis represents the top-1 accuracy.

## 3.1 CLASSIFICATION

It is well known that transformers are data-hungry which make them hard to train, mostly, due to the lack of the typical convolutional inductive bias. Consequently, the common protocol for self-supervised learning with transformers is to pretrain the model on a large scale dataset, such as ImageNet or even larger datasets. The compute and data demand of the vision transformers limit their adoption, particularly by AI researchers with smaller resource budget. Therefore, in the first set of experiments we investigate the applicability of training transformers from scratch with limited data. Particularly, we compare our proposed GMML approach with the state-of-the-art SSL methods when the pretraining and fine-tuning are performed only on the target dataset. Table 1 shows that our method outperforms the state-of-the-art with a large margin with an improvement of +1.3%, +17.4%, +23.2%, +35.2%, and +23.7% on Flowers, Pets, CUB, Aircraft, and Cars datasets, respectively.

Moreover, we show that longer pretraining (training for 3000 epochs) tends to achieve better performance rates with an improvement of +9.2%, +11.9%, +4.9%, +6.7%, and +2.6% on the aforementioned datasets. Figure 3 shows the increase in the top1-accuracy when the models are pretrained for longer which is an evident that GMML greatly benefits from longer pretraining where the performance is steadily improving even after 3000 epochs of pretraining.

Additionally, in order to study the effectiveness of GMML on bigger models, we pretrain GMML employing ViT-S variant of vision transformers for 3000 epochs. As shown in Table 1, we find that using a bigger transformer for self-supervised pretraining using GMML further improves the accuracy with an improvement of +4.1%, +2.1%, +6.2%, +0.4%, and +0.4% on the aforementioned datasets, respectively, compared to pretraining on ViT-T variant of transformers. Further, GMML significantly outperforms MAE [5] with a large margin on small datasets with an improvement of +7.6%, +15.0%, +18.0%, +15.5%, and +2.1% on the aforementioned datasets, respectively. Note that, for a fair comparison with MAE, we pretrained MAE for twice number of epochs as compared to GMML, i.e. 6000 epochs. We attribute the poor performance of MAE on small datasets to the

lack of inductive bias in the encoder part as the modelling of the local visual structures in MAE is only considered in the decoder part which is excluded during the finetuning phase.

### 3.1.1 TRANSFER LEARNING

After demonstrating the applicability of training transformers from scratch with limited data, we study the transfer ability of the representations learnt using GMML. In Table 2 and 3, we report the top-1 accuracy of the cross domain experiments employing ViT-T and ViT-S. Particularly, the on-diagonal cells indicate the performance when the models are pretrained and finetuned on the same dataset and the off-diagonal cells evaluate transfer performance across different datasets.

We observe that the proposed approach generalise well across different datasets even if the pretrained dataset and the target dataset are not from the same domain, e.g. CUB and Cars. This is attributed to the fact that GMML approach leverages unlabelled data in a task-agnostic way during the pretraining stage, hence the representations are not directly tailored to a specific classification task.

The second observation is that the number of images in the pretrained dataset matters. The more data the model sees during the pretraining, the better the accuracy, except for MNIST dataset.

MNIST is a toy dataset which has only 10 concepts, i.e. the digits, without any sort of variations in the background. In fact, it was expected that the pretrained model on MNIST dataset would not transfer well to other datasets. Yet, we note that the performance of the pretrained model on MNIST is much better than the performance when the model is trained from scratch with an improvement of 16.7%, 36.1%, 28.5%, 42.6%, and 57.9% on Flowers, Pets, CUB, Aircraft, and Cars datasets, respectively. These results demonstrate that pretraining the model with GMML mitigate the vision transformer's lack of inductive bias issue. More analysis and visualisation are in Appendix E.

Table 2: Domain Transfer. Fine-tuning self-supervised pretrained models employing ViT-T.

| Pretrain | Fine-tuning | | | | | |
|---|---|---|---|---|---|---|
| | MNIST | Flowers | Pets | CUB | Aircraft | Cars |
| random init. | – | 58.1 | 31.8 | 23.8 | 14.6 | 12.3 |
| *Transfer learning from toy dataset.* | | | | | | |
| MNIST | 99.6 | 74.8 | 67.9 | 52.3 | 57.2 | 70.2 |
| *Transfer learning from small datasets.* | | | | | | |
| Flowers | 99.6 | 90.6 | 78.7 | 61.8 | 67.4 | 80.2 |
| Pets | 99.5 | 88.8 | 86.0 | 61.7 | 69.1 | 82.7 |
| CUB | 99.5 | 89.1 | 84.8 | 71.2 | 77.79 | 88.7 |
| Aircraft | 99.5 | 89.2 | 84.4 | 68.7 | 85.1 | 89.7 |
| Cars | 99.6 | 89.2 | 85.7 | 69.4 | 81.1 | 92.7 |

Table 3: Domain Transfer (ViT-S).

| Pretrain | Fine-tuning | | | | |
|---|---|---|---|---|---|
| | Flowers | Pets | CUB | Aircraft | Cars |
| Flowers | 94.5 | 84.4 | 67.8 | 74.5 | 89.7 |
| Pets | 93.1 | 88.1 | 73.6 | 78.1 | 90.4 |
| CUB | 93.9 | 87.7 | 77.4 | 80.0 | 90.4 |
| Aircraft | 93.2 | 86.5 | 72.1 | 84.5 | 90.0 |
| Cars | 94.1 | 88.7 | 72.3 | 81.1 | 93.1 |

Table 4: Transfer Learning from ImageNet-1K employing ViT-T. * is reported by IDMM [30].

| Pretraining | Fine-tuning | | | | | |
|---|---|---|---|---|---|---|
| | Flowers | Pets | CUB | Aircraft | Cars | ImageNet-1K |
| *Training using only the given dataset* | | | | | | |
| Random Init* | 58.1 | 31.8 | 23.8 | 14.6 | 12.3 | – |
| Self-Supervised (GMML) | 90.4 | 86.0 | 71.2 | 84.1 | 92.7 | 76.4 |
| *Transfer learning from ImageNet-1K.* | | | | | | |
| Supervised (DeiT) [14] | 97.3* | 88.6* | 76.8* | 78.7* | 90.3* | 72.2 |
| Self-Supervised (GMML) | 97.9 | 89.2 | 81.9 | 86.2 | 93.2 | 76.4 |

Further, we show the benefits of transfer learning from large-scale dataset like ImageNet-1K. As shown in Table 4, pre-training the model in self-supervised fashion using GMML on ImageNet-1K outperforms supervised pre-training with a large margin, with an improvement of +0.6% +0.6% +5.1% +7.5% +2.9% +4.2% on Flowers, Pets, CUB, Aircraft, Cars, and datasets, respectively. An important characteristic of GMML is the ability of training transformers from scratch on the tiny datasets. This is reflected in Table 4. Notice, the reduction in performance gap between GMML pretraining on Imagenet-1K and GMML pretraining on the tiny dataset itself without any external information. Particularly, the gap on Aircraft and Cars reduces to around 1% between GMML based Imagenet-1K pretraining and the GMML pretraining on the dataset itself without any external knowledge.

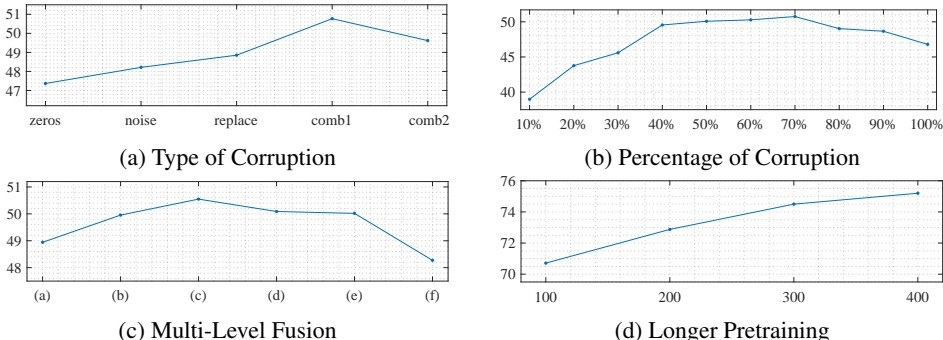

(a) Type of Corruption

(b) Percentage of Corruption

(c) Multi-Level Fusion

(d) Longer Pretraining

Figure 4: Ablation studies on the different recipes of GMML. The y-axis represents the top-1 accuracy on the ImageNet-1K validation set.

## 3.2 ABLATION STUDIES

For the ablation studies, all the models are pretrained on 5% of ImageNet-1K for 400 epochs and then finetuned for 400 epochs employing the small variant of vision transformer (ViT-S/16) [14]. The performance is assessed on the full validation set of ImageNet-1K. To set a baseline, we trained ViT-S/16 on the 5% of ImageNet-1K from scratch for 800 epochs where we obtained 30.38% top-1 accuracy on the validation set of ImageNet-1K. The poor performance is expected and in line with the previous observations, i.e. the performance of the pure transformer structure is poor when trained on small datasets from scratch due to the vision transformer's lack of inductive bias [11]. In the following, several ablation studies are conducted to investigate the different recipes of GMML.

**Effect of Type of Image Corruption (Alien Concepts).** In this set of experiments, we investigate the effect of different types of image corruption during the pretraining step. We start with vanilla transformers autoencoder where the model is pretrained as an autoencoder to reconstruct the input image, i.e. $D(E(x)) = x$, where $x$ is the input image, $E$ is the encoder which is ViT-S/16 in our case, and $D$ is a lightweight reconstruction decoder.

By visualizing the reconstructed images during the pretraining step, we found that the model is able to perfectly reconstruct the input image after few training epochs. Expectedly, after finetuning, the performance was similar to the performance of a model trained from scratch. Indeed, this is attributed to the fact that without proper choice of constraints, autoencoders are capable of learning identity mapping, i.e. memorising the input without learning any useful discriminative features.

To regularise the transformer-based autoencoder, we investigate the effect of applying different types of image inpainting including the following: randomly replacing group of connected patches from the image with zeros, noise, or replacing the patches with patches from another image. Further, we performed two more experiments with a combination of noise and replace (i.e. comb1 in Figure 4a) and a combination of zeros, noise, and replace (i.e. comb2 in Figure 4a). We corrupt upto 70% of the input image in the case of "zeros" and "noise" and upto 35% in the case of "replace" during the pretraining step. From Figure 4a, we found that the best individual inpainting task is "replace". Further, we obtained better performance when "replace" is combined with "noise" (i.e. "comb1"). On the other hand, the accuracy dropped when "zeros", "noise", and "replace" are combined together (i.e. "comb2"). The drop in performance might be attributed to the overly-constrained network, and reducing the drop percentage might help. We left the investigation of this point to the future work.

**The Effect of the extent of Image Corruption.** In this set of experiments, we show the impact of the masking ratio during the pretraining step. Figure 4b shows the top-1 validation accuracy when pretraining the ViT-S/16 with different levels of corruption percentages from upto 10% to upto 100% corruption per image. We found that the optimal ratio for vision is between 40% to 70% which is much higher than the masking ratio for NLP tasks [13], i.e. 15%. The masking encourages the network to learn semantic information from the uncorrupted patches surrounding the groups of masked tokens in order to recover the missing information where high masking ratio is required to challenge the model to learn useful salient features.

**The effect of Multi-level Feature Fusion for Reconstruction.** In addition to feeding the feature maps from the last block of ViT to the reconstruction head, we also investigated fusion of feature

maps from different blocks of the transformers before feeding them to reconstruction head. We investigated simple fusion by addition of the feature maps from different blocks and leave more complex fusion strategies for further studies. Note that fusion of features for reconstruction is only used during the pretraining stage of the network. Figure 4c shows the accuracy for different fusion settings. Setting (a) consists of combining the features maps from all the even blocks, i.e. 2, 4, 6, 8 , 10 and 12. Setting (b) combines blocks 4, 6, 8, 10, and 12. Setting (c) combines blocks 6, 8, 10, and 12 (d) combines blocks 8, 10, and 12. Setting (e) combines blocks 10 and 12. Lastly, setting (f) shows the original reconstruction effect from block 12 only.

We observe that feature fusion of blocks 6, 8, 10 and 12 gives the best results. This is inline with the visualisation shown in Figure 5. We note that the first four to six blocks of the transformer are used for modelling of introduced alien concepts. Therefore, adding features maps from first four blocks may not be beneficial.

**The Effect of Longer Pretraining.** As shown in Figure 4d, pretraining vision transformers with GMML leads to systematic performance gains in image classification.

## 4 DISCUSSION

GMML is a self-supervised learning mechanism for pretraining vision transformers with the ability to extract the information present in all the concepts in an image. GMML achieves this by manipulating randomly groups of connected tokens, contiguously covering a meaningful part of a semantic concept, and then recovering the hidden semantic information from the visible part of the concept. Unlike leading SSL approaches, GMML does not suffer from trivial solution, hence, it does not require tricky implementation mechanisms, which are commonly associated with modern SSL approaches. The transformers are unable to match the performance of CNNs on small and medium scale datasets due to the lack of so called inductive bias and require pretraining on huge datasets. GMML alleviate the problem of inductive bias by introducing, modelling and suppressing the alien concepts by local as well global context (see suplementary material for more detail). Hence, GMML based pretraining makes the vision transformers data-efficient. To our knowledge GMML is the first self-supervised pretraining work which consistently outperformed supervised pretraining for any pretraining and finetuning dataset, regardless of their sizes.

**Limitations:** Even though GMML establishes itself as state-of-the-art SSL and outperforms supervised pretraining with significant margin, it is merely a step towards the bigger goal of unsupervised semantic understanding of visual representation learning. Although GMML is aware of different concepts, it does not build an explicit representation for each concept in an image.

## 5 CONCLUSION

In this work we presented a self-supervised vision transformer, trained with unlabelled data to perform pre-text tasks. It is used as an autoencoder, exhibiting innovative architectural features comprising a light two-layer decoder, with a nonlinearity and transposed convolution to return the image representation back to the image space. The autoencoder enables the transformer to be trained using a Group Mask Machine Learning (GMML) strategy, which is instrumental in modelling contextual information present in all the concepts in the image. The GMML training involves corrupting each training image, and then attempting to reconstruct it from its visible parts. A reconstruction loss function is used to guide the learning process. GMML implicitly introduces a novel data augmentation technique. The key impact of the proposed GMML is that it makes it possible for transformers to train on small and medium size datasets. It is not only data efficient, but its outstanding information extraction ability enables it to outperform state-of-the-art supervised and self-supervised methods with large margins. The additional advantages include the simplicity and elegance of training, without the need to use large batches, momentum encoders, gradient stopping and other tricks to avoid solution collapse. GMML is currently the best mechanism to extract information from a given dataset and instil this information into transformer's weights. The source code will be made publicly available for the community to train on bigger corpora.

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

## A   APPENDIX

## B   COMPARISON WITH POST ART

There are several methods which have adopted the principles outlined in GMML at the beginning of 2021. In this section we briefly introduce these methods and discuss their similarities with and differences from GMML.

The two notable post arts are SimMIM [4] and MAE [5]. Similar to GMML both SimMIM and MAE use the principle of transformer based masked autoencoder. Both of them mask a high proportion of data-tokens randomly. However, we note that masking a very high proportion of data-tokens essentially defines groups of connected tokens. As can be seen in the ablation studies, their optimal masking proportion is very similar to GMML. Therefore, we can consider SimMIM and MAE as essentially variants/subsets of GMML, evaluated on a large dataset using large and huge vision transformers (ViT) models. The typical masking strategy used by SimMIM and MAE is masking by zero, GMML, in addition, uses random noise, as well more structured alien visual concept randomly sampled from another image in the batch. The ablation in Section 3.2 showed on 5% of Imagenet-1K that using a combination of random noise and structured alien concepts randomly gives better performance. Other than the masking strategy, SimMIM has minimal difference from GMML as they allow interaction between masked tokens from the beginning and use light decoder similar to GMML. MAE has two implementation differences which are interaction between tokens and so called decoder in transformers. MAE does not allow interaction between masked tokens for a number of layers in transformers. This allows faster processing due to less numbers of token during the multihead self attention. However, observing the number of epochs for pretraining between SimMIM and MAE, it is evident that SimMIM is converging faster. Both SimMIM and MAE used

ViT-B [11] model for pretraining using Imagenet-1K and finetuned on Imagenet-1K using classification labels. SimMIM achieved 83.8% by pretraining for 800 epochs while MAE obtained marginally lower performance of 83.6% while requiring twice as many epochs [1]. Another difference is the so called decoder for transformers. MAE emphasise that a slightly more complex decoder is needed for reconstruction. While SimMIM demonstrated that simple decoder as proposed by GMML is enough for pretraining transformers without supervision. In fact SimMIM, while deploying simple decoder, marginally outperformed MAE.

Another notable method in post art is BeIT [1]. BeIT uses external knowledge captured by an encoder trained without supervision, to group visual patches in order to define a visual vocabulary. This enables the use of cross entropy as a loss function, like in BERT [13]. However, unlike BERT the classes are coming from the external knowledge source, albeit trained without supervision. This can be considered as an expensive and extreme case of patch level distillation assisted by a supervised or unsupervised encoder, which is expensive. In addition, the approach will inevitably inherit issues of visual vocabulary ( a fixed number of visual words), a quantisation issue, visual ambiguity when assigning to cluster centres etc. Both SIMMIM and MAE demonstrate that the GMML via a masked autoencoder outperforms BeIT, which is hindered by these limitations.

Two notable extensions of GMML are MC-SSL [2] and iBOT [3]. Both are generalisations of the notion of GMML to non-autoencoder based learning tasks and achieved remarkable performance. MC-SSL in particular is attempting to make a step from contextual learning towards semantic learning.

## C  Experimental Details

The common evaluation to demonstrate the generalisation of the learnt features by self-supervised methods is to pretrain the model in an unsupervised fashion, followed by fine-tuning the model on a downstream task like image classification, object detection, segmentation, etc. In this work, we conduct several experiments on 7 well-known multi-class datasets (Table 5) to show the effectiveness of our proposed self-supervised vision transformer. In Section C.1, we provide the implementation details of the proposed GMML self-supervised training approach.

Table 5: Statistics of the employed datasets.

| Dataset | # Classes | #Training Samples | # Testing Samples |
|---|---|---|---|
| | Multi-class datasets | | |
| MNIST [33] | 10 | 60,000 | 10,000 |
| Flowers [34] | 102 | 2040 | 6149 |
| Pets [35] | 37 | 3680 | 3669 |
| CUB200 [36] | 200 | 5994 | 5794 |
| Aircraft [37] | 100 | 6667 | 3333 |
| Cars [38] | 196 | 8144 | 8041 |
| ImageNet-1K[6] | 1000 | 1.28M | 50,000 |

### C.1  Implementation Details

In our experiments, we implement the self-supervised architecture using the ViT transformer [11]. We employed the Tiny (ViT-T) and Small (ViT-S) variants of ViT with $224 \times 224$ input image size, $16 \times 16$ patch size, 12 consecutive MSA and MLP blocks. ViT-T and ViT-S have 192 and 384 hidden dimensions and 3 and 6 heads on each multi-head self-attention layer, respectively.

For the optimisation of the self-supervised training, the model is trained using the Adam optimiser [39] with a momentum of 0.9. The weight decay follows a cosine schedule [40] from 0.04 to

---

[1]Due to resource limitation we did not conduct a comparative study and we understand there can be implementation differences, hyper parameter selection, training strategies and other factors which may have contributed to faster convergence of SimMIM. Therefore, we just note the faster convergence of SimMIM and do not assert this finding

0.4, and the base learning rate is $5e^{-4}$. All the models are trained employing 4 Nvidia Tesla V100 32GB GPU cards with 64 batch size per GPU.

During the self-supervised training, simple data augmentation techniques are applied. We found that to learn low-level features as well as higher-level semantic information, aggressive data augmentation like MixUp [41] and Auto-Augment [42] hurts the training. Therefore, we used only cropping, colour jittering, as well as horizontal flipping by selecting a random patch from the image and resizing it to $224 \times 224$ with a random horizontal flip.

The augmented image is then corrupted using GMML based image manipulation. Specifically, we randomly either replace 70% of the patches with noise or 30% of the patches with structured alien concept, i.e. patches from another image.

For the finetuning step, the class head is embedded to the transformer with an output layer of $c$ nodes corresponding to the number of classes in the task in the downstream task. The model is optimised following the protocol used in Touvron et al. [14]. For the data augmentation, we applied random cropping, random horizontal flipping, MixUp and Auto-Augment during training.

## D  RECONSTRUCTION VISUALISATION

Figure 5 shows the reconstruction visualisation from different blocks corresponding to random noise alien concepts and visually structured alien concepts. We note that the first four to six blocks of the transformer are used for modelling of introduced alien concepts. Intermediate blocks model the contextual information present in different concepts and last blocks maybe used for refining the reconstruction [2]. Also note that GMML uses more transformers blocks when it comes to modelling of information which is introduced by visually structured alien concepts.

## E  INDUCTIVE BIAS

It is known that the vision transformers lack the explicit inductive bias in modelling the local visual structure [43]. This lack of inductive bias is modelled by training vision transformers on huge datasets [11]. From Figure 5 we note that the native concepts from the image are gradually diffused into the area which are manipulated by introduction of alien concepts. This may reflect the introduction of inductive bias by the GMML.

### E.1  GMML VS MAE

This experiment was to investigate the transfer ability of MAE pre-trained on small datasets. The comparison between GMML and MAE is shown in Table 6.

Table 6: Domain Transfer. Fine-tuning self-supervised pretrained models on different datasets employing ViT-S.

| Pretraining | Method | Fine-tuning | | | | |
|---|---|---|---|---|---|---|
| | | Flowers | Pets | CUB | Aircraft | Cars |
| Flowers | MAE | 86.87 | 58.81 | 42.28 | 37.74 | 46.72 |
| | GMML | 94.52 | 84.37 | 67.84 | 74.52 | 89.68 |
| Pets | MAE | 86.97 | 73.01 | 52.11 | 49.32 | 61.68 |
| | GMML | 93.06 | 88.09 | 73.59 | 78.13 | 90.43 |
| CUB | MAE | 88.32 | 71.89 | 59.35 | 56.97 | 70.34 |
| | GMML | 93.85 | 87.74 | 77.44 | 80.01 | 90.41 |
| Aircraft | MAE | 86.03 | 69.77 | 54.69 | 69.03 | 76.94 |
| | GMML | 93.15 | 86.54 | 72.11 | 84.52 | 90.01 |
| Cars | MAE | 92.29 | 81.00 | 65.03 | 78.46 | 91.03 |
| | GMML | 94.06 | 88.74 | 72.32 | 81.07 | 93.10 |

---

[2]The detail analysis of role of different blocks in GMML based self-supervised transformers will be presented in another study

| Original Image | Alien Concept | Rec. Block 1 | Rec. Block 2 | Rec. Block 4 | Rec. Block 6 | Rec. Block 8 | Rec. Block 10 | Rec. Block 12 |
|---|---|---|---|---|---|---|---|---|

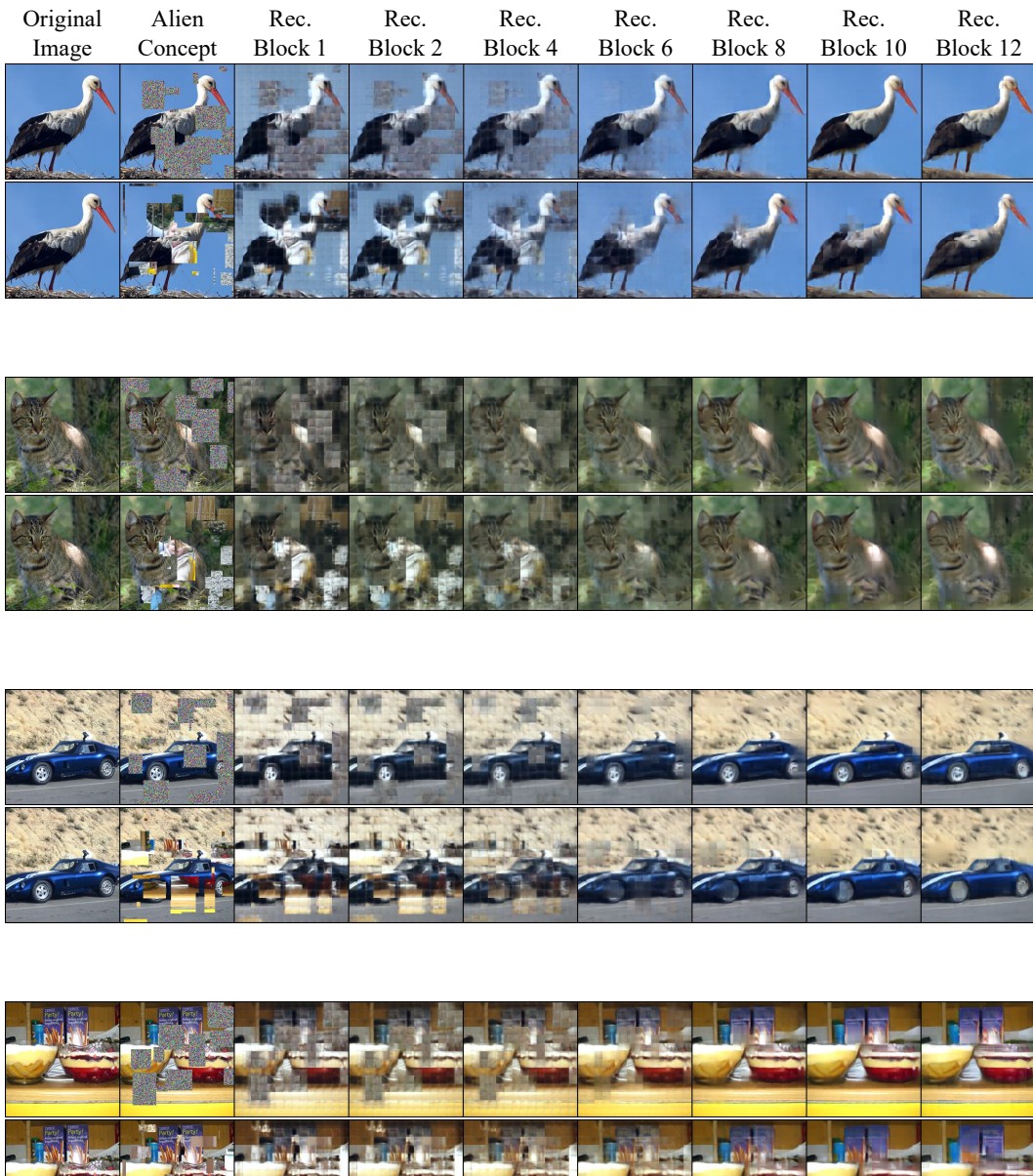

Figure 5: Reconstructed images from different transformer blocks after GMML-based pre-training employing two different alien concepts, Noise and Replace. For simplicity, we only corrupt the images by 30%.

While MAE is extremely powerful and has high generalisation ability when pre-trained on large-scale dataset like ImageNet-1K, it significantly suffers from generalisation ability when pretrained on small datasets, unlike GMML. After visualising the attention of the last block (Figure 7, we found that, unlike GMML, MAE does not overcome the inductive bias issue of data hungry transformers when pre-trained on small datasets.

## E.2  VISUALISATION

In Figure **??**, we show the visualisation of learnt self-attentions after GMML-based pretraining. Please note the block diagonal dominant lines which reflect the induction on induction bias specially in the shallow blocks of transformers.

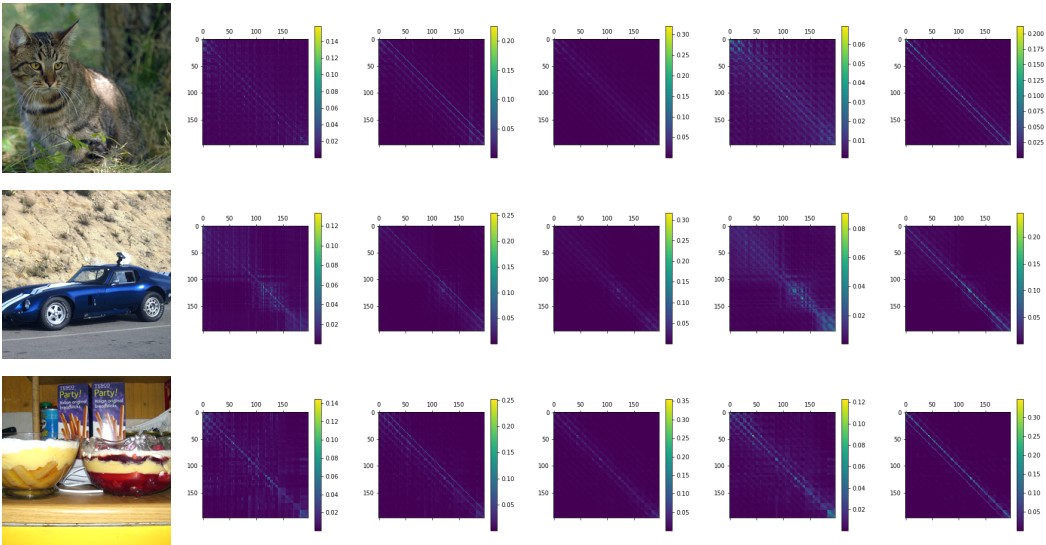

Figure 6: Visualization of average attention from block 2, 4, 6, 8, and 10 after GMML pretraining on 5% of ImageNet-1K.

We also show the attention visualisation of random samples after pretraining MAE on small datasets (Figure 7) and large-scale dataset, i.e. ImageNet-1k (Figure 8). MAE is able to overcome the induction bias issue of transformers only when pretrained on large-scale dataset.

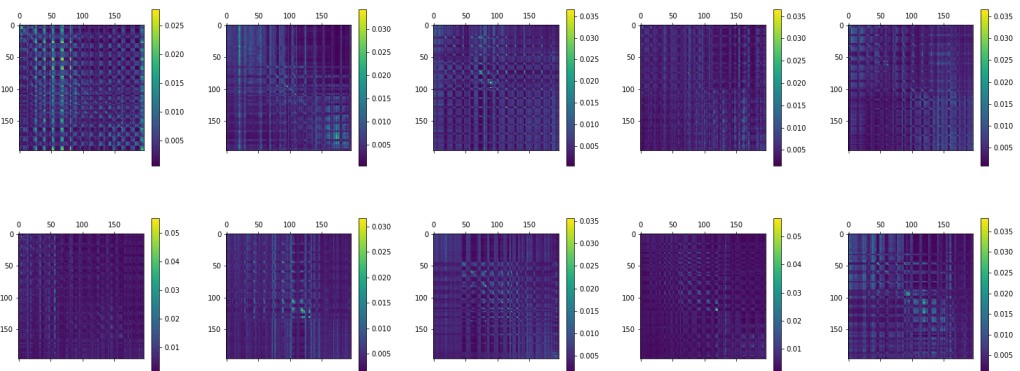

Figure 7: Relation between the tokens of randomly selected samples corresponding to the attention of the last block of the encoder of MAE pretrained on Pets dataset.

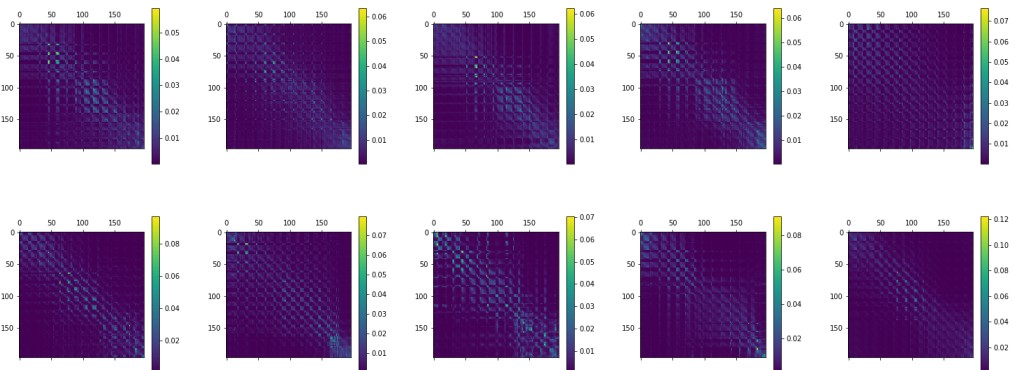

Figure 8: Relation between the tokens of randomly selected samples corresponding to the attention of the last block of the encoder of MAE pretrained on ImageNet-1K dataset.

