# OpenReview forum: "GMML is All you Need"
_ICLR.cc/2023/Conference — Submitted to ICLR 2023_

### Official Review · Reviewer_8Ka6 · 2022-10-24

**Confidence:** 4
**Correctness:** 3
**Technical Novelty And Significance:** 2
**Empirical Novelty And Significance:** 2
**Recommendation:** 3

**Clarity, Quality, Novelty And Reproducibility:**

Paper quite clearly written.
Not much issues with the quality of presentation.

Eqn. (1) does not appear to be novel;
No other analytics provided for strengthening the proposed concept - convergence, impact of bias,..etc

Experimentations appear to be reproducible, although not much time to explore myself
and verify the same.

**Strength And Weaknesses:**

Pros:
Transformers are hard to train, and providing any means to make them work for SSL
tasks, for DA/TL jobs is even harder.
Authors appears to address that problem to some extent,
Numerous results have shown the potential.

Cons
Although convinced by elaborate experimentations and ablations studies,
its difficult to get convinced due to lack of sufficient analytical proofs and justifications.

The manuscript has just one Eqn - the loss function used - with just the L1 vs L2 norm being discussed.
What about use of other loss functions - KLD and variants, Softmax, CE with variants etc. ?






**Summary Of The Paper:**

This paper proposes a Group Masked Model Learning (GMML), a Self-Supervised Learning (SSL) mechanism
for pretraining vision transformers with the ability to extract the contextual information
present in all the concepts in an image. This is achieved by manipulating
randomly groups of connected tokens, ensuingly covering a meaningful part of
a semantic concept, and then recovering the hidden semantic information from
the visible part of the concept. The paper suggests that GMML implicitly introduces a novel data augmentation
process.

Experimental results are shown for Image Classification, DA in Transfer Learning, and a few types of ablations studies.

At the end of the manuscript, the authors claim the following -
GMML based pretraining makes the vision transformers data-efficient. Also, GMML is the first
self-supervised pretraining work which consistently outperformed supervised pretraining for any
pretraining and finetuning dataset, regardless of their sizes.

The key impact of the proposed GMML is that it makes it possible for transformers
to train on small and medium size datasets. It is not only data efficient, but its outstanding information
extraction ability enables it to outperform state-of-the-art supervised and self-supervised
methods with large margins.


**Summary Of The Review:**


I am not convinced with the paper.
There appears to be some contribution - but not substantial.


Note for Area Chairs and PC chairs:
This paper is available in
https://arxiv.org/abs/2205.14986
in which the author names are revealed.

The authors of above are a subset of those in [2] in this submitted manuscript.

---

### Official Review · Reviewer_rp7V · 2022-10-24

**Confidence:** 2
**Correctness:** 2
**Technical Novelty And Significance:** 2
**Empirical Novelty And Significance:** 2
**Recommendation:** 5

**Clarity, Quality, Novelty And Reproducibility:**

-

**Strength And Weaknesses:**

Strength:
The work reports promising results for self supervised learning in low data regime.



Weaknesses:
1. The novelty of the work is confusing as the work looks quite similar to other popular approaches for self supervised learning (such as MAE). However, the authors actually claim that the already published MAE (and also SIMMIM) extended actually the proposed work.

2. There are other strong frameworks for self supervised learning that are not referred to in this work, for instance, data2vec (Baevski, Alexei, et al. "Data2vec: A general framework for self-supervised learning in speech, vision and language." arXiv preprint arXiv:2202.03555 (2022).)

3. It is difficult to understand the contribution of the proposed approach on larger datasets. For instance, how this work results are compared with the state-of-the-art on the full ImageNet. Particularly, where the proposed approach fits in the Table 1 of the  data2vec paper.

4. Some parts are unclear, for instance, why the imput image is corrupted up to 70% in the case of “zeros” and “noise”, while in the case of “replace” it is corrupted up to 35%. Also not sure why the combination of “replace” and “noise” works better.


**Summary Of The Paper:**

The work proposes Group Masked Model Learning framework for self supervised learning.

**Summary Of The Review:**

-

---

### Official Review · Reviewer_4sPn · 2022-10-25

**Confidence:** 5
**Correctness:** 2
**Technical Novelty And Significance:** 2
**Empirical Novelty And Significance:** 2
**Recommendation:** 3

**Clarity, Quality, Novelty And Reproducibility:**

- (Clarity) I couldn't quite understand why the GMML model presented in 2021 should be a contribution to this paper. (see C2, C5)
- (Quality) The experiment presented in this paper is only a classification experiment on small-scale datasets, and it is too insufficient to claim the title "GMML is All you Need". (see C1) Even in the experiments performed, there are papers omitted. (see C3)
- (Novelty) I consider this model to be about the same as SimMIM.
- (Reproducibility) Source code is not provided, but implementation details in the appendix provide reproducibility.

**Strength And Weaknesses:**

- (C1) Models those capable of dealing with small-scale datasets are always welcome. However, assessing the model’s performance on large-scale datasets (at least ImageNet-1K) is needed to determine whether the model can also be used for pre-training those datasets. In that sense, the paper has its weakness in only delivering experiments on small-scale datasets and small-sized models.
- (C2) GMML is almost identical to SimMIM. The authors go on to argue that structures like SimMIM all stem from the GMML structures presented in 2021, but it is questionable why this paper submitted to ICLR 2023 should take the contribution of GMML presented in 2021. Since SimMIM is already presented in CVPR 2022, it is appropriate to address that this paper is conveying the variant of SimMIM, regardless of the existence of GMML architecture, that I failed to find its reference in the paper.
- (C3) Table 1 in the paper omits several papers, including SimMIM [1], BEiT [2], and SplitMask [3]. Especially, SplitMask shares topics with this paper: “SSL method that is capable of being pre-trained on small-scale dataset” and reports small datasets’ metrics seemingly on par with this paper’s result.
- (C4) The authors said that most non-GMML approaches suffer from trivial constant solutions, but the representation collapsing problem only occurs for positive (BYOL-like) methods. Contrastive learning does not suffer from the collapse by design and the number of works adopting contrastive learning is not neglectable.
- (C5) The manuscript has a self-contradicting statement that they have already suggested GMML in 2021 and picked a proposal of GMML as one of the paper's contributions.

[1] Xie, Zhenda, et al. "Simmim: A simple framework for masked image modeling." *Proceedings of the IEEE/CVF Conference on Computer Vision and Pattern Recognition*. 2022.
[2] Bao, Hangbo, et al. "BEiT: BERT Pre-Training of Image Transformers." *International Conference on Learning Representations*. 2021.
[3] El-Nouby, Alaaeldin, et al. "Are Large-scale Datasets Necessary for Self-Supervised Pre-training?." *arXiv preprint arXiv:2112.10740* (2021).

**Summary Of The Paper:**

This paper delivers a method called group mask model learning (GMML). GMML learns a representation of an image by partially corrupting it and reconstructing it with a transformer encoder-decoder structure. The authors conducted experiments that show GMML’s superiority in learning representations from small-scale datasets with small-sized models.

**Summary Of The Review:**

Due to the various weaknesses described above (see C1-C5) I cannot give a high recommendation to this paper.

---

### Official Review · Reviewer_JV27 · 2022-10-25

**Confidence:** 4
**Correctness:** 2
**Technical Novelty And Significance:** 2
**Empirical Novelty And Significance:** 2
**Recommendation:** 3

**Clarity, Quality, Novelty And Reproducibility:**


The authors claim that GMML is proposed “at the beginning of 2021” with no citation. They also claim that SimMIM and MAE are extensions of GMML. So, I have no choice but to search for GMML and found an unpublished paper on ArXiv [1].

Ignore [1], in my opinion, the novelty of this paper mainly lies in using alien patches when masking inputs.

[1] SiT: Self-supervised vIsion Transformer. 2104.03602

**Strength And Weaknesses:**

Strength

1. The benefit from using noise or alien patches from other images, rather than just zeros when masking inputs is clear as shown in Figure 4(a).
2. On small datasets, GMML converge faster and perform better (Table 1)

Weakness

There are several critical weaknesses in experiments.

1. Firstly, in the experiments, the authors only use ViT-tiny and ViT-small backbones. While it is more common to use a larger backbone for comparison (e.g. B, L, H) as in MAE and SimMIM.
2. Secondly, the authors missed the comparison of ImageNet-1k in Table 1, which is probably the most important comparison with other baselines.
3. Thirdly, some important baselines are missing for comparison, e.g. SimMIM, BEIT

**Summary Of The Paper:**

This paper proposes a self supervised learning framework GMML for vision transformers. The idea is essentially the same as MAE and SimMIM (the authors claim that MAE and SimMIM are extensions of GMML). When masking the inputs, they use noise or alien patches from other images, rather than just zeros. In this way, the initial blocks of the networks have to model the alien and the non-alien concepts, and thus can better model the context.

**Summary Of The Review:**

Overall, the paper propose an interesting masking method using alien patches. However, the experiments are weak and the relationship with MAE and SimMIM is unclear.

---

### Author Response · Authors · 2022-11-18
**Comments to all the reviewers**

First, we would like to thank the reviewers for their comments, and we are happy to see that learned reviewers have made a critical point that SIMMIM is **almost identical** to GMML, and MAE and other latest works are **very similar** to GMML, which is a message that we have been trying to convey for more than a year despite its originality and merits which to date are valid.

Our main aim is to set the record straight in this open review process. Thus, we are not going to address one by one, but instead focus on the underlaying trend of all the reviews.

The innovations of GMML (Group Masked Model Learning) for vision transformers, although the origins of the method go back to April 2021, differentiate it from all the image masking-based self-supervised learning methods, which appeared subsequently. Many of the features are still novel in comparison with the current state of the art. Note that the original SiT paper was the first to make self-supervised learning work with transformers, by showing the extent of masking needed (in comparison with language) to make it work. The proposed GMML

- is masking by corrupting the image with the content of another image.

- is unique by using nonaligned masking.

- is using a combination of masking signals (zero, noise, replacement).

- In contrast to MAE, our masking involves a group of connected unmasked tokens, which helps to learn basic image properties.

- Our contribution includes a clear experimental evidence that our proposed method is better on small datasets than all the existing competitors.

- We show the merits of GMML on large scale dataset, i.e. ImageNet-1k, on tiny vision transformers.

- We have furnished extensive evidence from domain transfer experiments that the quality of the learnt representation by GMML is significantly better than that provided by any of the self-supervised learning methods that emerged after April 2021.

These features are clearly stated in the paper.

The impressive performance of GMML on small datasets was acknowledged by the reviewers (thank you very much). However, the key criticism, which precipitated rejection, appears to be that GMML was not tested on large datasets and large backbone architectures. We evaluated on large scale dataset, i.e., ImageNet-1K using ViT-Tiny. However, we did not evaluate on large scale dataset employing large scale models. First of all, such testing can be afforded mainly by AI giants, as it requires computational resources well beyond those available to typical university groups. For an average researcher, this is certainly not feasible, and making it a pre-condition of acceptance deliberately stifles a major segment of the research community. Most importantly, the argument that any method has to be demonstrated on large datasets and architectures is flawed. We are not saying that because GMML is better on small datasets, it must be better on large datasets as well by extrapolation. Such claims would need a proof. What we are saying and experimentally proving is that on small datasets our method is better than all the self-supervised learning methods that are top ranking on large datasets. This is valuable information for many in the community who work on applications involving small datasets.

---

### Decision · Program_Chairs · 2023-01-20

**Decision:**

Reject

**Justification For Why Not Higher Score:**

Many specific questions and comparisons raised by the reviewers are not addressed in the author response.

**Justification For Why Not Lower Score:**

N/A

**Metareview: Summary, Strengths And Weaknesses:**

*Summary*: The paper proposes a framework for self-supervised learning that uses image masking and reconstruction. The authors propose to corrupt the input image using zero, noise patches or patches from different images. The method is evaluated on image classification benchmarks.

*Strengths*: (1) The method learns by masking the input which is a technique used in other domains such as text. (2) The masking strategy proposed in this work is different from other work such as BEiT, MAE, SimMIM, iBOT whereby the authors use zero, noise patches or patches from other images. (3) The proposed method works well on smaller network architectures and datasets.

*Weaknesses*: (1) The main weakness raised by all the reviewers is the similarity of several published works from 2022 (SimMIM, SplitMask, MAE, BEiT, data2vec) and this work. While the authors argue that the original GMML predates these methods, the AC does not believe that the argument is relevant to the current paper (which by the authors own presentation is an improvement over the GMML work from 2021). Thus, as noted by every other reviewer, there are critical missing experimental comparisions. (2) The author response is incomplete as it misses specific questions raised by the reviewers (about loss function, analysis, comparisons). I do see that the authors address questions raised about the scale of the experiments which is useful context.